# Newborn Screening for Inborn Errors of Metabolism by Next-Generation Sequencing Combined with Tandem Mass Spectrometry

**DOI:** 10.3390/ijns10020028

**Published:** 2024-03-29

**Authors:** Chengfang Tang, Lixin Li, Ting Chen, Yulin Li, Bo Zhu, Yinhong Zhang, Yifan Yin, Xiulian Liu, Cidan Huang, Jingkun Miao, Baosheng Zhu, Xiaohua Wang, Hui Zou, Lianshu Han, Jizhen Feng, Yonglan Huang

**Affiliations:** 1Department of Guangzhou Newborn Screening Center, Guangzhou Women and Children’s Medical Center, Guangzhou Medical University, Guangdong Provincial Clinical Research Center for Child Health, Guangzhou 510180, China; fangfang_violet@sina.cn; 2Department of Genetic, Shijiazhuang Maternal and Child Health Hospital, Shijiazhuang 050090, China; 13933119153@163.com; 3Department of Pediatric Endocrinology and Genetic Metabolism, Shanghai Institute for Pediatric Research, Xinhua Hospital, Shanghai Jiaotong University School of Medicine, Shanghai 200092, China; chenting02@xinhuamed.com.cn; 4Neonatal Disease Screening Center, Jinan Maternity and Child Health Hospital Affiliated to Shandong First Medical University, Jinan 250001, China; liyulinchange@163.com (Y.L.); zouhui819@163.com (H.Z.); 5Department of Genetics, Inner Mongolia Maternity and Child Health Care Hospital, Hohhot 750306, China; 15024920955@163.com (B.Z.); wangxiaohua2222@163.com (X.W.); 6Department of Medical Genetics, NHC Key Laboratory of Preconception Health Birth in Western China, Yunnan Provincial Key Laboratory for Birth Defects and Genetic Diseases, Yunnan Provincial Clinical Research Center for Birth Defects and Rare Diseases, The First People’s Hospital of Yunnan Province/The Affiliated Hospital of Kunming University of Science and Technology, Kunming 650032, China; zyh8920002@163.com (Y.Z.); bszhu@aliyun.com (B.Z.); 7Department of Pediatrics, Chongqing Health Center for Women and Children &Women and Children’s Hospital of Chongqing Medical University, Chongqing 401147, China; xxkk27403@gmail.com (Y.Y.); jennamiao@aliyun.com (J.M.); 8Neonatal Disease Screening Center, Hainan Women and Children’s Medical Center, Haikou 570206, China; lxl507@163.com (X.L.); 13807652278@139.com (C.H.)

**Keywords:** inborn errors of metabolism, newborn screening, next-generation sequencing, tandem mass spectrometry

## Abstract

The aim of this study was to observe the outcomes of newborn screening (NBS) in a certain population by using next-generation sequencing (NGS) as a first-tier screening test combined with tandem mass spectrometry (MS/MS). We performed a multicenter study of 29,601 newborns from eight screening centers with NBS via NGS combined with MS/MS. A custom-designed panel targeting the coding region of the 142 genes of 128 inborn errors of metabolism (IEMs) was applied as a first-tier screening test, and expanded NBS using MS/MS was executed simultaneously. In total, 52 genes associated with the 38 IEMs screened by MS/MS were analyzed. The NBS performance of these two methods was analyzed and compared respectively. A total of 23 IEMs were diagnosed via NGS combined with MS/MS. The incidence of IEMs was approximately 1 in 1287. Within separate statistical analyses, the positive predictive value (PPV) for MS/MS was 5.29%, and the sensitivity was 91.3%. However, for genetic screening alone, the PPV for NGS was 70.83%, with 73.91% sensitivity. The three most common IEMs were methylmalonic academia (MMA), primary carnitine deficiency (PCD) and phenylketonuria (PKU). The five genes with the most common carrier frequencies were *PAH* (1:42), *PRODH* (1:51), *MMACHC* (1:52), *SLC25A13* (1:55) and *SLC22A5* (1:63). Our study showed that NBS combined with NGS and MS/MS improves the performance of screening methods, optimizes the process, and provides accurate diagnoses.

## 1. Introduction

Newborn screening (NBS) for inborn errors of metabolism (IEMs) is a successful public health project for detecting life-threatening or long-term health conditions to reduce morbidity and mortality [1]. Tandem mass spectrometry (MS/MS) is a key technique for accessing the NBS program that allows for the rapid detection of many metabolites in dried blood spots (DBS). Measurement of amino acids and acylcarnitines enables the identification of approximately 40 to 50 IEMs a few days after birth. In general, expanded NBS using MS/MS has been widely used worldwide with several advantages, including rapid and convenient procedures, significantly increased detection of IEMs [2], early diagnosis and recognition of death [3] and cost effectiveness [4]. However, the positive predictive value (PPV) is low, and the results may be nonspecific, with additional false positives [5], and poor sensitivity, with additional false negatives [5]. Moreover, when multiple diseases are screened via single-tier analysis of one marker, MS/MS cannot distinguish among different diseases and subtypes, which is not conducive to timely accurate diagnosis and treatment [6].

Second-tier tests, secondary biomarkers, postanalytical tools (R4S and CLIR) [7,8] and random forest (RF) machine learning [9] have been used to improve the performance of NBS via MS/MS. False-positive rates can indeed be reduced, but performance improvements are still limited [10]. Confirmation of a biochemical diagnosis is usually achieved by identifying pathogenic genetic variants; this diagnostic trajectory is often a protracted and lengthy process resulting in delays in diagnosis, and importantly, therapeutic intervention for these rare conditions is also postponed [11].

With advances in next-generation sequencing(NGS) technologies, an increasing number of genetic diseases of unknown cause with nonspecific presentations are best diagnosed by exome sequencing (ES) or NGS [12]. Moreover, evidence demonstrates that ES and NGS are reliable for detecting pathogenic variants in genomic material extracted from DBS [13,14]. Furthermore, confirmatory or second-tier testing using ES/NGS may improve the performance of NBS [15,16]. Newborn genetic screening has also been proven to be successful when using single monogenetic disease or target genetic sequencing panels [17]. Nevertheless, the National Bureau of California (NBSeq) project showed that ES alone was insufficiently sensitive or specific to be a primary screening tool for most NBS IEMs [18]. Thus, we investigated the outcomes of NBS in a certain population via NGS via a multicenter approach in which a gene panel was used for a general newborn population as a first-tier screening test [19] combined with MS/MS.

## 2. Materials and Methods

### 2.1. Recruitment and Sample Collection

A total of 29,601 newborn born from February 2021 to December 2021 were enrolled from 8 newborn screening centers located in the east, west, south, and north of China, including Shanghai Xinhua Hospital (SH), Chongqing Maternal and Child Health Care Hospital (CQ), Guangzhou Women and Children’s Medical Center (GZ), First People’s Hospital of Yunnan Province (YN), Hainan Women and Children’s Medical Center (HN), Jinan Maternal and Child Health Care Hospital (JN), Shijiazhuang Maternal and Child Health Care Hospital (SJZ), and Inner Mongolia Maternal and Child Health Care Hospital (NMG). These centers were selected to represent the nationwide population. DBS samples (at least four 8 × 8 mm pieces) were collected between 3 and 7 days after birth, dried at room temperature, and stored at 4 °C. The amount of DBS in each specimen was tested for genetic NBS and expanded NBS by MS/MS simultaneously. Informed consent was obtained from all participants, and approval was obtained from the institutional review board of each center (XHEC-C-2020-054-1). The project has been registered in the Chinese Clinical Trial Registry (ChiCTR2100043025) and the National Health Security Information Platform Medical research registration information system (MR-44-21-016154).

### 2.2. Panel of Screened Disorders

For MS/MS screening, the PE kit nonderivative method (PerkinElmer, Waltham, MA, USA) was used to assess 38 amino acid disorders/organic acid disorders/fatty acid oxidation disorders (AAs/OAs/FAODs), including 15 amino acid disorders (AAs), 11 organic acid disorders (OAs), and 12 fatty acid oxidation disorders (FAODs). For genetic screening, a custom-designed panel targeting the coding region of the 142 genes (128 diseases) (Integrated DNA Technologies, Coralville, IA, USA) was applied. The library was sequenced with a minimum depth of 100× coverage on a genetic sequencer (MGI). An in-house verified variant calling pipeline was used to analyze single nucleotide variants, small insertions and deletions, and copy number variants (CNV) including CNVs involving 2 or more continuous exons in DMD, exon 7 deletion of SMN1, and common CNVs involving HBA1/HBA2/HBB [19]. A total of 52 genes associated with the 38 IEMs were tested by MS/MS and are summarized in Appendix A.

### 2.3. Expended NBSs by MS/MS

DBS samples from 29,601 newborns were analyzed via the nonderivative method (Neobase TM) with MS/MS at the newborn screening centers. In total, 11 amino acids, 30 acylcarnitines, free carnitine, and succinylacetone were tested with regard to 38 kinds of IEMs (Appendix A). Every disease had two or more indicators, including metabolites and ratios. Different newborn screening centers have different reference cutoff values, and a screening-positive result was considered according to the reference cutoff value of each screening center, and the patient was recalled for review.

### 2.4. NBS by NGS

DBS samples from 29,601 newborns were submitted to the clinical laboratories of the Beijing Genomics Institute for genetic sequencing. Genomic DNA was extracted from DBS using a MagPure Tissue DNA KF Kit (Magen, Guangzhou, Guangdong, China). DNA fragmentation, end repair, and 3′ end tailing were performed using a VAHTS6 Universal Plus Fragmentation Module (Vazyme, Nanjing, Jiangsu, China). A custom-designed panel targeting the coding region of the 142 genes (128 diseases) (Integrated DNA Technologies, Coralville, IA, USA) was applied. Consensus about the criteria for variant prioritization and suspicious positive criteria for genetic screening was reached in multicenter studies [19]. A screening-positive result for a gene was defined as a pathogenic (P) or likely pathogenic (LP) variant with a matching inheritance pattern without knowledge of the phenotype and clinical features of the tested individual. Patients with positive results and their families were recalled for confirmatory testing.

### 2.5. Diagnosis

High-risk infants whose screening was positive by either base-NGS or base-MS/MS screening were referred for confirmatory tests and follow-up. The confirmatory tests varied for different diseases and included blood biochemical tests such as complete blood count, kidney function, liver function, ammonia, lactic acid, homocysteine, and urine organic acids and family inheritance validation, etc. Diagnostic decisions were based on genotype, MS/MS results, and confirmatory tests.

### 2.6. Statistical Analysis

We present the characteristics of the study participants as a whole and separately by MS/MS and genetic screening. For categorical data, summary data are reported as frequencies and percentages. Statistical analysis was performed using SPSS 17.0 software (SPSS, Inc., Chicago, IL, USA).

## 3. Results

### 3.1. Performance of NBS by NGS Combined with MS/MS

Among 29,601 newborns, 23 cases (6 with MMA (5 cblC-MMA, 1 cblA-MMA); 5 with PCD; 3 with PKU; 2 with SCADD; 2 with IBD; 1 with NICCD; 1 with MCAD; 1 with CPTII; 1 with maple syrup urine disease, MSUD; 1 with MADD) of neonatal IEMs involving amino acid, organic acid, or fatty acid metabolism disorders were diagnosed (Table 1). The incidence of IEMs was approximately 1 in 1287 patients.

As shown in Table 2, the performances of genetic screening and MS/MS screening were compared and analyzed among 29,601 neonates from participating centers.

### 3.2. Assessment of NGS as a First-Tier Screening Test

For genetic screening alone and without MS/MS synchronization screening, 24 (0.08%) patients were positive, and 24 (100%) patients were recalled for review. Seventeen IEMs were diagnosed; there were seven false positives. Of these seven patients, three had complex heterozygous variants, one had a homozygous variant in the *PAH* gene, one had complex heterozygous variation in the *PRODH* gene, one had complex heterozygous variation in the *ACADSB* gene, and the last one, who is a male, had a hemizygote variation in the *OTC* gene with an X-linked disorder. All these variations were P or LP variants and matched the family inheritance pattern when these patients were recalled for confirmatory tests. However, all of them were unaffected based on normal biochemical results (Appendix A).

Six cases would have been missed but with the results of MS/MS screening and the recall review received positive results. These six cases had MSUD, IBD, PCD, SCADD, MADD, and CPTII, respectively. The PPV of genetic screening was 70.83%, and the sensitivity was 73.91%.

### 3.3. Assessment of MS-MS as a First-Tier Screening Test

For MS/MS screening alone, 507 (1.71%) patients were positive, and 397 (78.3%) were recalled for review. In total, 21 patients with IEMs were diagnosed without genetic synchronization screening. There were 376 false positives, and all of these patients were recalled for diagnosis or censoring but had normal results.

Two patients were missed but had positive genetic screening results: one with NICCD and another with cblC-MMA. The PPV of MS/MS screening was 5.29%, and the sensitivity was 91.3%.

### 3.4. Biochemical and Molecular Characteristics of the Diagnosed Patients

Eleven different genes were found among the 23 diagnosed patients, including *MMACHC*, *MMAA*, *ACAD8*, *ACADS*, *ACADM*, *PAH*, *SLC22A5*, *SLC25A13*, *CPT2*, *DBT*, and *ETFA*, all exhibited recessive inheritance, involving 34 different variants (Table 3). Among the 23 patients diagnosed by genetic screening combined with MS/MS, 17 (P1–P17) were genetic-screening positive with positive or negative MS/MS results, and 6 (P18–P23) were genetic-screening negative with positive MS/MS results. In one of these six patients, P23, the detected *ETFA* gene was not included in the gene panel; the others were detected to carry at least one variant of uncertain significance (VUS). Two patients (P5 and P17) were MS/MS-screening negative with positive genetic results. One patient diagnosed with cblC-MMA (P5) was compound heterozygous for two LP missense variants (c.482G>A and c.565C>T) of the *MMACHC* gene with a normal propionylcarnitine (C3) level but abnormal confirmatory test results during the recall review. Another patient diagnosed with NICCD (P17) was compound heterozygous for two P variants (c.1638_1660dup and c.852_855delTATG) of the *SLC25A13* gene with a normal citrulline (Cit) level at the first screening but an increased Cit level at reexamination. Of these six confirmed cases of MMA, four (P2–P5) had normal propionylcarnitine (C3) levels, but three (P2–P4) had true-positive MS/MS results indicating that methionine (Met) was decreased or that its ratio with acetylcarnitine (C3/C2) was increased. Moreover, confirmatory test results showed that methylmalonic acid in urine (27.09–38.06 mmol/mol creatinine) and blood serum homocysteine (31.8–123.9 µmol/L) were increased. Notably, all three of these patients were cblC-deficient and harbored c.80A>G complex heterozygous variants of the *MMACHC* gene. Of the five confirmed cases of PCD, three carried c.1400C>G complex heterozygous variants, and one had a c.1400C>G homozygous variant of the *SLC22A5* gene.

### 3.5. Carrier Frequencies and Geographical Distribution of Targeted Genes

The carrier frequencies of variants in 52 genes were calculated (Appendix A). The five genes related to the highest variant carrier frequencies in these newborns were *PAH* (1:42), *PRODH* (1:51), *MMACHC* (1:52), *SLC25A13* (1:55) and *SLC22A5* (1:63). Variants of the *PAH* and *MMACHC* genes had higher carriage rates in the northern participating units NMG, JN, and SJZ, which was consistent with the regional distribution of confirmed cases. *SLC25A13* and *SLC22A5* had higher carriage rates in the southern participating units GZ and HN, which was also consistent with the geographical distribution of confirmed cases. The composition and geographical distribution of the variants of these four genes in the healthy population were determined (Appendix A). The top ten variants of these four genes are summarized in Figure 1. These three variants of the *PAH* gene that were the most common in the healthy population were c.728G>A, c.721C>T, and c.611A>G. The top three variants of the *MMACHC* gene were c.609G>A, c.658_660delAAG, and c.482G>A. The variants c.852_855delTATG/c.790G>A were found to be *SLC25A13* gene hotspots. The variants c.1400C>G and c.51C>G were *SLC22A5* gene hotspots. In addition, the genes with a carrier rate greater than 1:550 were MADD (*ETFDH*), SCADD (*ACADS*), BH4D (*PTS*), IBD (*ACAD8*), MCAD (*ACADM*), VLCAD (*ACADVL*), BTD (*BTD*), 3-MCC (*MCCC1*, *MCCC2*), CACT (*SLC25A20*), CTLN1 (*ASS1*), PA (*PCCB*), and HHHS (*SLC25A15*).

## 4. Discussion

Genetic NBS has been proven to be successful when used to evaluate monogenic diseases in China. In combination, genetic NBS and traditional NBS can complement each other [20]. This is the latest, prospective pilot study of NBS using NGS combined with MS/MS results for IEMs.

Our results showed that genetic screening improved the specificity and sensitivity of MS/MS screening. After MS/MS screening alone and subsequent confirmation via genomic sequencing, the PPV was 5.29%, and the number of false-positive patients was 376. However, for NGS, a first-tier newborn screening test, the PPV reached 70.83%, which means that there were fewer false positives and fewer unnecessary follow-up visits [5]. In this study, although we conducted genetic screening combined with MS/MS screening and recalled both genetic and MS/MS screening positive-patients simultaneously, it seems that the number of false positives in this study did not decrease. However, if combined screening is routine in the future, these MS/MS screening false-positive cases, especially those that are boundary and negative with genetic screening, should not be recalled. The sensitivity ranged from 91.3% to 100%. The two missed patients were MS-MS-screening negative but were revealed by genetic screening to be NICCD and cblC-MMA, the two most common diseases identified during newborn screening in China [21]. Elevated citrulline levels and several citrulline-based ratios are key indices of NICCD screening by MS/MS, but these indices are not very sensitive because citrulline levels might not increase immediately after birth [22,23]. Even at elevated levels, MS/MS could not determine whether the patient had NICCD or type I citrullinemia. The MMA concentration was identified by MS-MS using specific cutoff values for propionylcarnitine (C3) and its ratio with acetylcarnitine (C2). This patient (P5) with cblC-MMA was missed because her C3 level and C3/C2 were normal but with compound heterozygous for two missense variants (c.482G>A/c.565C>T) of the *MMACHC* gene; the variant c.482G>A is associated with milder disease [24] and late onset [25]. Therefore, these two missing patients were very difficult to avoid by MS/MS. In addition to the patient (P5), three patients (P2–P4) with cblC deficiency had normal C3 levels. However, they were defined as being true-positive cases by MS/MS screening only for high C3/C2 or low Met levels (Table 3). Each screening center may have its own interpretation criteria, and most screening centers will follow the main indicator of C3 increase before considering this ratio. The judgment of negative or positive involves more the interpretation standards of different screening centers. Thus, the combined screening method can optimize processes for rapid diagnosis.

Well-recognized genotype-phenotype correlations are helpful for the diagnosis, treatment, and follow-up of diseases. All three of the patients harbored the c.80A>G complex heterozygous variant of the *MMACHC* gene. The variant c.80A>G is reported to be associated with late-onset diffuse lung disease (DLD) [26] and prominent renal complications, and the most frequent renal pathological manifestation is thrombotic microangiopathy [27]. The variant c.80A>G should be considered to indicate disease of the cardiovascular system [25]. Analysis of the composition of the *MMACHC* gene variants in the healthy population (Appendix A) revealed the four most common variants to be c.609G>A, c.658_660delAAG, c.482G>A, and c.80A>G, which is basically consistent with reported data for Chinese patients. The most common c.609G>A variant is reported to likely lead to early-onset cblC disease [28]. However, patients with the c.80A>G and c.609G>A compound heterozygotic variant genotypes had late-onset disease with neuropsychiatric symptoms and pulmonary hypertension [25]. In this study, two patients (P2 and P3) had the same genotype. In addition, the spread of the c.80A>G and c.609G>A variants in Chinese patients has been reported to be caused by a founder effect [29].

Although seven patients were defined as having false-positive results in genetic screening (Appendix A), they met the genetic diagnosis criteria but had biochemically normal or borderline cutoff values and did not have any clinical symptoms. Among the four patients with variants in the *PAH* gene, one had the c.516G>T homozygous variant, one had the c.516G>T compound heterozygous variant associated with mild PKU [30], one had the c.1068C>A compound heterozygous variant, and one had the c.728G>A compound heterozygous variant; these two variants are clearly associated with classic PKU [31]. One patient with c.1322T>C/c.273+1G>C in the *PRODH* gene had the same genotypes as the reported patient, with no clinical manifestations observed at 5 years or 7 months [32]. The other patient was a male with suspected OTCD and the c.830 G>A variant in the *OTC* gene was found in one patient with late-disease onset [33]. X-linked ornithine transcarbamylase deficiency (OTCD), the most common urea cycle defect [34], is difficult to detect by expanded NBS due to the insensitivity of the citrulline indicator [35]. A patient is defined as positive for genetic screening and negative for MS/MS with a normal citrulline level in the early stage. The onset of OTCD symptoms is extremely variable [36]. Considering the possibility of late onset, long-term attention and follow-up are needed for these patients, which is a major challenge for genetic screening. At the same time, for NGS, the epigenetics of some diseases should also be considered [37]. However, due to the fact that this study is only an NGS sequencing panel of genes, it is limited.

Combined with MS-MS screening, the sensitivity of genetic screening was improved by this approach. In our study, genetic screening alone, without knowledge of the phenotype and clinical features of the tested individuals, was insufficiently sensitive (73.91%) as a first-tier screening test, with a sensitivity lower than that of MS/MS (91.3%). This finding is comparable with the reference sensitivity of 88% for ES compared to 99.0% for MS/MS [18]. Based on biochemical positive results, except for one patient (P23) for whom the genetic method was not used, five genetic screening false-negative cases were considered positive in reanalysis, which elevated the sensitivity to 95.6% (22/23). Moreover, MS-MS screening supplements the deficiencies of genetic screening. The case (P23) with the *ETFA* gene not included in the gene panel was detected by MS-MS screening and then diagnosed by ES, which emphasizes the importance of genetic selection in genetic screening [20].

Due to the small sample size, the epidemiological incidence rate could not be statistically analyzed. However, according to the healthy population carrying rate and Hardy Weinberg’s law [38], the population disease incidence rate could be estimated. The highest incidence of genetic metabolic disease screening by MS/MS was detected in the HPA population in China [21], which was consistent with the highest carrier frequency of *PAH* (up to 1/42) in this study. The prevalence of MMA, PCD, and NICCD, which are associated with a high incidence of disease in China [21], was also consistent with the carrier rate. Moreover, we clearly delineated the geographical distribution of diseases and genes. PKU and MMA are the two most common disorders in the northwestern Chinese population [39,40], and PCD is the most common disorder in the southern Chinese population [41,42]. In this study, the carriage rates of *PAH*, *MMACHC*, *SLC25A13*, and *SLC22A5* gene variants were consistent with the regional distribution of confirmed cases. Appendix A shows that the carrier rate of the *SLC25A20* variant was the highest in GZ. GZ accounted for the largest proportion of the hotspots of the c.199-10T>G variant of *SLC25A20* in the healthy population (Appendix A), which was related to the founder effect [43] and explained by the incidence of CACT being reportedly higher in the Guangdong area than in the other regions [3]. For the same reason, the hotspots *PAH* and *MMACHC* in the healthy population were distributed mainly in JN and SJZ, and the hotspots of *SLC25A13* and *SLC25A5* in the healthy population were distributed mainly in GZ (Figure 1).

In conclusion, NBS-based NGS combined with MS/MS significantly improved the sensitivity and specificity, optimized the process, and provided accurate diagnoses. With general support for genetic screening [44] and the guidance of expert consensuses about the application of NGS in the screening of monogenic diseases [20], it is believed that genetic screening combined with MS/MS can better serve in NBS for IEMs. The cohort of NBS could be implemented for future recommendations.

## Figures and Tables

**Figure 1 IJNS-10-00028-f001:**
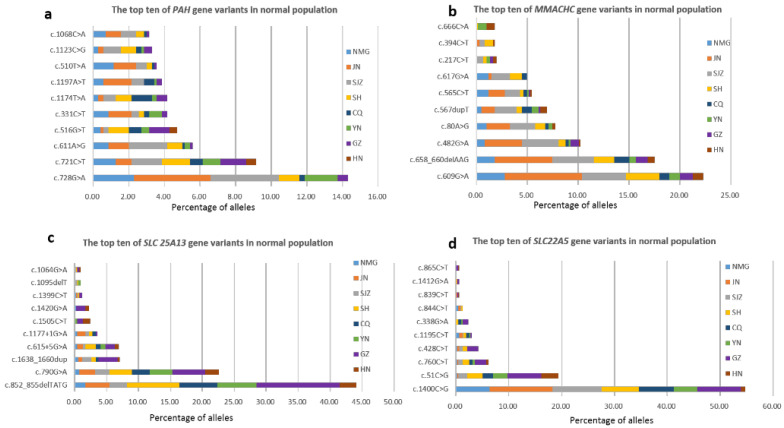
The top ten of four genes’ variants in normal population and geographical distribution in this multicenter study. (**a**) The top ten *PAH* gene variants. (**b**) The top ten *MMACHC* gene variants. (**c**) The top ten *SLC 25A13* gene variants. (**d**) The top ten *SLC22A5* gene variants.

**Table 1 IJNS-10-00028-t001:** Incidence and carrier frequencies of inborn errors of metabolism detected by next-generation sequencing combined with tandem mass spectrometry among 29,601 newborns.

Disease Name	Gene	Diagnosed Cases	Incidence Rate	Carrier Frequencies
cblC-MMA	*MMACHC*	5	1/5920	1/52
PCD	*SLC22A5*	5	1/5920	1/63
PKU	*PAH*	3	1/9867	1/42
SCADD	*ACADS*	2	1/14801	1/127
IBD	*ACAD8*	2	1/14801	1/211
NICCD	*SLC25A13*	1	1/29,601	1/55
MCAD	*ACADM*	1	1/29,601	1/251
CPTII	*CPT2*	1	1/29,601	1/580
cblA-MMA	*MMAA*	1	1/29,601	1/1139
MSUD (type II)	*DBT*	1	1/29,601	1/1480
MADD	*ETFA* ^1^	1	1/29,601	/
Total		23	1/1287	

*ETFA* ^1^: This gene was not included in the NGS method used in this study.

**Table 2 IJNS-10-00028-t002:** Comparison of conventional newborn screening by tandem mass spectrometry using dried blood spot testing versus next-generation sequencing in a multicenter population of neonates.

MS/MS Screening								Genetic Screening					
Units	Neonates	Positive Cases	Number of Recalls (Recall Rate)	Confirmed Cases	PPV	False Negative Cases	Confirmed Disease	Positive Cases	Number of Recalls (Recall Rate)	Confirmed Cases	PPV	False Negative Cases	Confirmed Disease
SH	4888	47	41 (87.2%)	1	2.44	0	MMA	1	1 (100%)	1	100	0	MMA
GZ	4813	50	47 (94%)	4	8.51	1	MMA, PCD (2), MADD	5	5 (100%)	4	80	1	MMA, NICCD, PCD (2)
JN	4797	149	138 (92.6%)	6	4.35	0	MMA (2), IBD, CPTII, PCD, MCAD	6	6 (100%)	4	66.67	2	MMA (2), PCD, MCAD
SJZ	4899	67	65 (97.0%)	2	3.08	0	PKU, MSUD	1	1 (100%)	1	100	1	PKU
CQ	2988	30	22 (73.3%)	2	9.09	0	PKU, PCD	3	3 (100%)	1	33.33	1	PKU
YN	3006	43	40 (93.0%)	2	5.00	1	SCADD, IBD	3	3 (100%)	3	100	0	MMA, SCADD, IBD
NMG	3233	113	36 (31.86%)	2	5.56	0	PKU, MMA	3	3 (100%)	2	66.67	0	PKU, MMA
HN	977	8	8 (100%)	2	25	0	PCD, SCADD	2	2 (100%)	1	50	1	PCD
Total	29601	507	397 (78.3%)	21	5.29	2	9 (kinds)	24	24 (100%)	17	70.83	6	7 (kinds)

**Table 3 IJNS-10-00028-t003:** Diagnosis of inborn errors of metabolism using next-generation sequencing combined with tandem mass spectrometry.

Diagnosed Cases No.	Genetic/MS-MS Screening	Primary Screening Results (Cutoff)	Recall Review Results (Cutoff)	Genes	Exon	Nucleotide Change	Protein Change	ACMG	Zygosity	Disorders	Mode of Inheritance
								Category			
P1	+/+	C3 15.19 (5)	9.01 (5.5)	*MMAA*	EX2	c.365T>C	p.Leu122Pro	LP	Hom	cblA-MMA	AR
P2′	+/+	C3 4.4 (4.5),	C3 4.37 (4.5),	*MMACHC*	EX1	c.80A>G	p.Gln27Arg	LP	Het	cblC-MMA	AR
		C3/C2 0.356 (0.2)	C3/C2 0.344 (0.2)	*MMACHC*	EX4E	c.482G>A	p.Arg161Gln	LP	Het		AR
P3	+/+	C3 3.53 (5), Met8.36 (9),	C3 0.92 (5), Met7.17 (9),	*MMACHC*	EX1	c.80A>G	p.Gln27Arg	LP	Het	cblC-MMA	AR
		C3/C2 0.18 (0.2)	C3/C2 0.25 (0.2)	*MMACHC*	EX4E	c.609G>A	p.Trp203 *	LP	Het		AR
P4	+/+	C3 2.23 (4.5),	C3 3.64 (4.5),	*MMACHC*	EX1	c.80A>G	p.Gln27Arg	LP	Het	cblC-MMA	AR
		C3/C2 0.235 (0.2)	C3/C2 0.44 (0.2)	*MMACHC*	EX4E	c.609G>A	p.Trp203 *	LP	Het		AR
P5 ^1^	+/-	C3 1.99 (3.59)	C3 2.16 (4.5)	*MMACHC*	EX4E	c.482G>A	p.Arg161Gln	LP	Het	cblC-MMA	AR
		C3/C2 0.14 (0.2)	C3/C2 0.11 (0.22)	*MMACHC*	EX4E	c.565C>T	p.Arg189Cys	LP	Het		AR
P6	+/+	C3 9.74 (4.8), C3/C2 0.57 (0.23)	C3 7.83 (4.8), C3/C2 0.96 (0.23)	*MMACHC*	EX4E	c.658_660delAAG	p.Lys220del	P	Hom	cblC-MMA	AR
P7	+/+	Phe 276 (116),	Phe 393 (116),	*PAH*	EX3	c.331C>T	(p.Arg111 *)	P	Het	PKU	AR
		Phe/Tyr 3.02 (1.5)	Phe/Tyr 4.57 (1.5)	*PAH*	IVS12	c.1315+6T>A		LP	Het		AR
P8	+/+	Phe 606 (120),	Phe 10.89 (mg/dL)	*PAH*	EX5	c.482T>C	p.Phe161Ser	LP	Het	PKU	AR
		Phe/Tyr 14.11 (1.5)		*PAH*	EX11	c.1197A>T	p.V399V	P	Het		AR
P9	+/+	Phe 2.45 (mg/dL) (2.1)	Phe 166 (120),	*PAH*	EX6	c.688G>A	p.Val230Ile	LP	Het	PKU	AR
			Phe/Tyr 1.88 (1.2)	*PAH*	EX12	c.1238G>C	p.Arg413Pro	P	Het		AR
P10	+/+	C0 6.78 (10)	C0 10.09 (10)	*SLC22A5*	EX1	c.51C>G	p.Phe17Leu	LP	Hom	PCD	AR
P11	+/+	C0 7.46 (10)	C0 4.64 (10)	*SLC22A5*	EX2	c.428C>T	p.Pro143Leu	P	Het	PCD	AR
				*SLC22A5*	EX8	c.1400C>G	p.Ser467Cys	P	Het		AR
P12	+/+	C0 8.82 (10)	C0 4.81 (10)	*SLC22A5*	EX4	c.760C>T	p.Arg254 *	P	Het	PCD	AR
				*SLC22A5*	EX8	c.1400C>G	p.Ser467Cys	P	Het		AR
P13	+/+	C0 5.16 (9)	C0 7.76 (9)	*SLC22A5*	EX8	c.1400C>G	p.Ser467Cys	P	Hom	PCD	AR
P14	+/+	C4 2.8 (0.46) C4/C3 3.79 (0.4)	C4 1.98 (0.46) C4/C3 6.04 (0.4)	*ACADS*	EX9	c.1031A>G	p.Glu344Gly	P	Hom	SCADD	AR
P15	+/+	C4 2.08 (0.46)	C4 1.99 (0.46)	*ACAD8*	EX3	c.289G>A	p.Gly97Arg	LP	Het	IBD	AR
		C4/C3 4.1 (0.4)	C4/C3 2.02 (0.4)	*ACAD8*	EX4	c.413delA	p.Asn138Metfs * 36	P	Het		AR
P16	+/+	C8 3.64 (0.13),	C8 2.16 (0.13),	*ACADM*	IVS10	c.946-1G>C		LP	Het	MCAD	AR
		C8/C1015.8 (1.4)	C8/C10 12 (1.4)	*ACADM*	EX11	c.1085G>A	p.Gly362Glu	LP	Het		AR
P17	+/-	Cit 20.13 (30)	Cit 449.85 (35)	*SLC25A13*	EX16	c.1638_1660dup	p.Ala554Glyfs * 17	P	Het	NICCD	AR
				*SLC25A13*	EX9	c.852_855delTATG	p.Met285Profs * 2	P	Het		AR
P18	-/+	Leu 1088 (270),	Leu 4182.3 (270),	*DBT*	EX2	c.75_76delAT	p.Cys26Trpfs * 2	P	Het	MSUD	AR
		Val 602 (269)	Val 998.6 (269)	*DBT*	EX11E	c.1359_1360delAG	p.Arg453Serfs * 3	VUS	Het		AR
P19	-/+	C4 1.19 (0.5)	C4 1.23 (0.5)	*ACAD8*	EX4	c.473A>G	p,Tyr158Cys	VUS	Het	IBD	AR
				*ACAD8*	EX10	c.1165C>T	p.Arg389Trp	VUS	Het		AR
P20	-/+	C4 0.9 (0.45)	C4 2.11 (0.45)	*ACADS*	EX3	c.322G>A	p.Gly108Ser	LP	Het	SCADD	AR
				*ACADS*	EX6	c.779G>T	p.Gly260Val	VUS	Het		AR
P21	-/+	C0 6.65 (9)	C0 8.96 (9)	*SLC22A5*	EX8	c.1400C>G	p.Ser467Cys	P	Het	PCD	AR
				*SLC22A5*	EX3	c.621G>T	p.Gln207His	VUS			AR
P22	-/+	C12 1.18 (0.3), C14 2.52 (0.4), C16 25.1 (4.27),	C12 0.9 (0.33), C14 0.57 (0.4),	*CPT2*	EX1	c.125C>T	p.Thr42IIe	VUS	Het	CPTII	AR
		C18 5.82 (1.8), C18:1 8.94 (3)	C18:1 4.24 (3)	*CPT2*	EX4	c.1613delA	p.Tyr538Serfs * 5	VUS	Het		AR
P23	-/+	C4 2.19 (0.8), C5 3.11 (0.35), C6 1.14 (0.12),	C4 5.83 (0.8), C5 6.2 (0.35), C6 3.15 (0.12),	*ETFA* ^1^	EX5	c.369G>A	p.R122K	P	Het	MADD	AR
		C8 1.76 (0.16)	C8 5.2(0.16)	*ETFA* ^1^	EX8	c.659delC	p.S220Lfs * 6	P	Het		AR

P5 ^1^: Although C3 and C3/C2 ratios were normal, urine organic acids and other findings were abnormal. +: denotes that the case was a true positive by genetic screening or by MS/MS screening. -: denotes that the case was a false positive by genetic screening or by MS/MS screening. *ETFA* ^1^: This gene was not included in the NGS method used in this study. *: means stop codon.

## Data Availability

Data will be made available to qualified researchers on request.

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
