# Peer review of "Newborn Screening for Inborn Errors of Metabolism by Next-Generation Sequencing Combined with Tandem Mass Spectrometry"

_2409-515X, 2024, doi:10.3390/ijns10020028_

Round 1

Reviewer 1 Report

Comments and Suggestions for Authors

The authors describe two approaches to newborn screening in 29,601 newborns during 2021 with patients recruited from 8 newborn screening centres in China.

They compared the effectiveness of both MS/MS and Exome sequencing using dried blood spot samples and reported both specificity and sensitivity of detection for 38 inherited metabolic disorders.

They show that the PPV% and sensitivity of detection using MS/MS was 5.3% and 91.3% respectively while for exome sequencing this was 70.8% and 73.9% respectively.

They conclude that the two used in combination may offer acceptable sensitivity while significantly increasing specificity (PPV%).

In general the paper is thoughtful and well written and the conclusions from the findings are appropriate.

The extensive use of poorly defined abbreviations throughout makes it difficult for the reader eg p1/2, line 51-52: PAH, PRODH, MMACHC, SLC25A and SLC22S5.    I managed to get to the end of the paper without ever understanding what the PRODH gene was.   This could be helped by a list of the abbreviations used.

On p3 line 147, it states that the ‘diagnostic decisions were based on genotype, clinical phenotype and expert consensus’.   It does not make it clear whether metabolite concentration was used to guide the decisions or whether the ‘biochemical’ phenotype was being used as a surrogate for ‘clinical’ phenotype.   The lack of clear case definitions, established at the outset ie before testing began, weakens the paper as opinions/consensus can be poorly defined.

On p5 Table 2, in the columns relating to MS/MS screening, it is unclear what the difference is between the ‘Recall positive cases’ and the ‘Confirmed cases’.   It is not easy to understand how a patient could be deemed a positive case following recall but remain unconfirmed?

On p9, line 320, the authors should probably acknowledge that the sensitivity offered by exome sequencing is less good than that available in whole genome sequencing, so the relatively poor sensitivity is not a limitation or indictment of a genomic approach but rather depends upon the method selected.

Although the authors do hint at this when describing some cases as ‘mild’, it may be useful to explain the biochemical results can inform not just the disease classification but also its severity – in that way these approaches are also complementary by offering some early indications of prognosis and the management/treatment required.

The paper is rather lengthy so anything that can be done to make it more concise may benefit the reader.

In general the use of English is excellent and this is a limited but thought provoking and topical paper which should be published after some limited revisions.

Reviewer 2 Report

Comments and Suggestions for Authors

Dear Authors,

this is a very important paper that share multidimensional experience and integration between geneomic and metabolomic. I want to point out the attention on the description in the abstract and in the text about MMA, could you oplease correct with MMA with homocystinuria because the term MMA is linked with a diffrent defect. The explanation and the discussion are well detailed and very useful appears the tables. I appreciate also the different assessment by NGS and MS/MS and the results appear very excellent with a great difference in term of PPV and sensitivity.

At pag 5, line 178 you write about OTC gene but I suggest to explain the peculiarity of transissione of this disorder (X-linked) to understand better the results. The delicate problem of NGS could be discuss more appropriately: the problem of epigenetic variant (es PRDX for CblC patients, see Cavicchi C. et al, Clin Epigenetics, 2021), the interpretation of VUS.

The cohort of NBS could be implemented for future reccomendations.

Clin Epigenetics  

Comments on the Quality of English Language

Good quality
